# Progressive Motor and Non-Motor Symptoms in *Park7* Knockout Zebrafish

**DOI:** 10.3390/ijms24076456

**Published:** 2023-03-29

**Authors:** Lakshmi Narasimha Murthy Chavali, Ingeborg Yddal, Ersilia Bifulco, Simen Mannsåker, Dagne Røise, Jack O. Law, Ann-Kristin Frøyset, Sushma N. Grellscheid, Kari E. Fladmark

**Affiliations:** 1Department of Biological Sciences, University of Bergen, 5020 Bergen, Norway; 2Computational Biology Unit, University of Bergen, 5020 Bergen, Norway

**Keywords:** *Park7*, DJ-1, zebrafish, Parkinson’s disease, behavior, NAD metabolism

## Abstract

DJ-1 is a redox sensitive protein with a wide range of functions related to oxidative stress protection. Mutations in the *park7* gene, which codes for DJ-1 are associated with early onset familial Parkinson’s disease and increased astrocytic DJ-1 levels are found in pathologic tissues from idiopathic Parkinson’s disease. We have previously established a DJ-1 knockout zebrafish line that developed normally, but with aging the DJ-1 null fish had a lowered level of tyrosine hydroxylase, respiratory mitochondrial failure and a lower body mass. Here we have examined the DJ-1 knockout from the early adult stage and show that loss of DJ-1 results in a progressive, age-dependent increase in both motoric and non-motoric symptoms associated to Parkinson’s disease. These changes coincide with changes in mitochondrial and mitochondrial associated proteins. Recent studies have suggested that a decline in NAD+ can contribute to Parkinson’s disease and that supplementation of NAD+ precursors may delay disease progression. We found that the brain NAD+/NADH ratio decreased in aging zebrafish but did not correlate with DJ-1 induced altered behavior. Differences were first observed at the late adult stage in which NAD+ and NADPH levels were decreased in DJ-1 knockouts. Considering the experimental power of zebrafish and the development of Parkinson’s disease-related symptoms in the DJ-1 null fish, this model can serve as a useful tool both to understand the progression of the disease and the effect of suggested treatments.

## 1. Introduction

Parkinson’s Disease (PD) is a progressive neurodegenerative disorder that affects nearly 4% of the population worldwide over age 65, making it the second-most common neurodegenerative disease [1]. The diagnosis of the disease is based on motor deficits as a result of a gradual loss of dopaminergic neurons in the *Substantia nigra*. However, non-motor symptoms such as anxiety and depression, in addition to altered inflammatory factors, precede the point of diagnosis [2]. Available treatments for PD do not have a significant effect on disease progression and are mainly directed toward motor symptoms. Therefore, there is a high demand to understand the molecular mechanisms underlying the disease and to differentiate stages of events into causes or consequences in order to develop novel treatments. In this respect, animal models are of utmost importance to facilitate the investigation of PD-relevant molecular deficits and analyze the effect of targeting them.

Loss-of-function mutations in the *park7* gene cause a rare form of familial PD [3]. *Park7* codes for the DJ-1 protein, which, through multiple actions protects cells from oxidative stress [4,5,6]. Increased astrocytic DJ-1 expression is found in post-mortem samples from sporadic PD patients and other neurodegenerative diseases [7], indicating that DJ-1 has a general role in protecting neurons from oxidative stress. Prolonged exposure to oxidative stress can, however, induce irreversible hyperoxidation of specific cysteines of DJ-1 rendering the protein in an inactive state [8].

Vertebrate knockout models of DJ-1 have been established in mice [9,10], rats [11,12], and zebrafish [13,14]. In contrast to the rat and zebrafish models, DJ-1 knockout mice retain normal nigrostriatal function. There is also a discrepancy in the mouse and rat models regarding whether the loss of DJ-1 induces progressive loss of motoric function or not [10,11,12].

Zebrafish models are emerging as a powerful tool for in vivo studies of human neurodegenerative diseases as they are relatively easy to genetically manipulate, have a high conservation of human disease genes, and share similarity in neuroanatomy. We have previously shown that proteins involved in inflammation, mitochondrial function, and oxidative stress protection are altered in young zebrafish lacking DJ-1 [13]. Late adults showed tyrosine hydroxylase reduction, dopamine downregulation, and weight loss. Using the zebrafish DJ-1 knockout as a PD model is particularly appealing as these early molecular changes and late features observed overlaps with aspects associated with PD pathogenesis [15,16,17].

It is generally accepted that mitochondrial dysfunction is an underlying factor in PD [18,19]. NAD is an essential cofactor in mitochondrial respiration, in which it is reversible and interconverted between its oxidized and reduced states. NAD is also the precursor of NADP, which has an important role in the defense against oxidative stress [20]. More recently, it has been suggested that increasing NAD+ levels may amend the effects of mitochondrial deficit [21] as dopaminergic neurons derived from human iPSC expressing PD-related gene mutations show a reduction in NAD levels and NAD+/NADH ratios [22,23].

Here we have analyzed PD-related motor and non-motor symptoms from an age point downstream of the previously observed molecular changes in mitochondrial and inflammatory factors [13]. We show that the PD-associated symptoms increase in an age-dependent manner. We also compare the NAD-metabolism in the aging brains of DJ-1 knockout and wild type animals and demonstrate an age-dependent decrease in NAD+/NADH ratios in both knockout and wild type, with a difference in NAD+ and NADPH levels first appearing at the late adult stage.

## 2. Results and Discussion

### 2.1. DJ-1 Knockouts Do Not Gain Weight after 6 Months of Age

We have previously observed that late adult DJ-1 knockout (DJ-1 KO) zebrafish weigh less compared to their wild type littermates [13]. Here, we wanted to examine at which age weight loss was pronounced. As shown in Figure 1 wild type fish increase their weight from 6 months and until 24 months, whilst the DJ-1KO did not gain weight after 6 months. A progressive weight loss due to the absence of DJ-1 was not observed. Similar results have been obtained in DJ-1 knockout mice from the age of 12 months [9]. Weight loss is among the non-motor symptoms associated to parkinsonism, and early weight loss predicts a poor outcome [24,25]. Although not proven, this weight loss is suggested to be a result of improper energy balance due to a dysregulated homeostasis and increased energy demand associated with Parkinson’s Disease (PD) symptoms. In mice, DJ-1 has been shown to have a role in regulating energy homeostasis and glucose balance by regulating brown adipose tissue [26]. DJ-1 knockout mice had lower body weight and were more insulin sensitive compared to wild types [26]. Since zebrafish do not have brown fat [27], the absence of weight gain in DJ-1 KO zebrafish cannot be explained similarly. It should be noted that we have previously observed a loss of mitochondrial complex I activity in muscle and indications of a metabolic switch from mitochondrial metabolism to glycolysis in our DJ-1 knockout zebrafish [13] which may be associated with the lower weight of DJ-1 KOs compared to wild type fish.

### 2.2. Loss of DJ-1 Induces Age-Dependent Progressive Motor and Anxiety Associated Behavior Deficits

PD is classically characterized by motor deficits such as tremors and bradykinesia [28]. However, anxiety is also common and adds to a number of non-motor symptoms of PD [29,30]. Thus, both are important and useful readouts when using PD animal models. We used the established “novel tank test” to measure these parameters in our DJ-1 knockout model [31].

Three age groups (6, 12, and 18 months) of DJ-1 knockouts and their wild type littermates were subjected to behavior analysis (Figure 2). The analysis included both motor function as determined by the distance moved (Figure 2A) and anxiety related behavior (Figure 2B,C) in a specific time frame.

Our results showed that already at 6 months of age DJ-1 KO had a significant decrease in mobility compared to age-matched wild type animals (Figure 2A). The reduced mobility was even more severe in older DJ-1 KO, whilst no significant changes were observed in wild type animals (Figure 2A). Similar age-dependent progressive reduction in locomotory activity has been observed in both rat [9,11] and mouse [10] DJ-1 knockouts. However, in the latter case there were no changes in the nigrostriatal function as have been observed in both rat and zebrafish DJ-1 knockouts [11,13]. Contradiction also appears in the effect of DJ-1 loss in rats as Sanchez-Catasus et al., did not observe any age-dependent reduction in motoric behavior in their rat DJ-1 knockout [12]. The discrepancy in the effects of DJ-1 loss on motor behavior and dopaminergic neurons in rodent models strengthens the value of the use of DJ-1 KO zebrafish as a PD model.

Anxiety related non-motoric behavior was assessed as time spent in the lower part of the novel tank (latency, Figure 2B) and total time without movement (freezing, Figure 2C) [32]. DJ-1 deficient zebrafish showed a tendency to spend longer time in the lower half of the novel tank, a behavior that became more prominent with age (Figure 2B and Appendix A). In their normal tank, however, their time spent in the upper half did not seem to differ to what was observed for WT (Appendix A). There was also a significant age-dependent increase in freezing duration in DJ-1 KO in contrast to wild type animals (Figure 2C).

The first appearing difference between DJ-1 KO and wild type in the novel tank test was the distance travelled (Figure 2A), which showed a significant difference already in the youngest age group. It is hard to define this solely as a motor symptom as it may also be a symptom of increased anxiety associated to reduced exploratory behavior.

Our behavior results are comparable to what has been observed in rodent models of PD after toxicant induced dopaminergic cell death [33]. Paraquat and 6-hydroxydopamine exposure induced both motor deficits and increased anxiety-related behavior in addition to selective dopaminergic neuronal degeneration. In contrast, DJ-1 null rats showed no age-dependent (6–9 months) anxiety behavior [11,12], although a decrease in exploration activity was observed [11]. The disparate results may be both due to differences in experimental protocols and the age at which the organism was tested. On the contrary, it is appealing that our DJ-1 knockout model exhibit both motor deficits and non-motor symptoms as related to PD.

### 2.3. NAD+/NADH Ratio Declines in Aging Zebrafish, but DJ-1 Knockout Only Affects NAD+ Levels at the Late Adult Stage

A decline in NAD+ levels is suggested as a hallmark of aging [34], and a negative correlation between NAD+ levels and age-associated oxidative stress has been shown [35]. Increasing NAD+ levels by supplementation of its precursors is neuroprotective and counteracts dopaminergic cell loss in *Drosophila* and *Caenorhabditis elegans* models of PD [36,37,38]. Thus, replenishment of NAD-levels is under consideration as a neuroprotective therapy for PD [39]. In view of the antioxidative properties of DJ-1 it was of interest to determine the effect of DJ-1 loss on NAD-metabolism.

We analyzed NAD metabolites using mass spectrometry from whole brain samples from DJ-1 KO and wild type animals at the age of 6, 10, and 24 months (Figure 3 and Figure 4). By investigating the NAD+/NADH ratio we were able to compare the different age-groups although they were not analyzed in the same mass spectrometry run. Even though there was an exceptionally high variation in NAD+/NADH ratios at 6 months of age in both DJ-1 KO and wild type brains a two-way ANOVA revealed that there was a statistically significant age-dependent reduction in NAD+/NADH ratios (F (2,33) = 14, *p* < 0.0001) which occurred between 6 and 10 months of age (Figure 3A). On the other hand, there was no significant effect of genotype (F (1,33) = 0.01, *p* = 0.9117). A difference between DJ-1 KO and wild type was first observed at 24 months when comparing NAD+ amount normalized with protein content (Figure 4A). As the three different age groups were not simultaneously analyzed we could not compare the relative amount between the different age groups. Neither could we conclude if the reduction in NAD+/NADH ratio is driven by a decrease in NAD+ levels. Importantly, our results show that changes in NAD-homeostasis do not correlate with behavior deficits in DJ-1 deficient animals and thus, cannot be the driving force behind PD-related symptoms.

NAD+ is essential in mitochondrial homeostasis both by its involvement in the TCA cycle as a central electron donor in oxidative phosphorylation and induction of mitophagy [21]. Mitochondrial dysfunction is associated with PD [40]. Also, DJ-1 is proposed to have an important role in mitochondrial homeostasis [41]. Therefore, one might expect to see a difference in NAD+/NADH ratios between DJ-1 KO and wild type which was not the case. It should be noted that we have analyzed whole brain and that differences might have appeared if we would have been able to analyze specific areas, e.g., *S. nigra*. An age-dependent decline in NAD+ levels has been observed in the hippocampus of aging mice, although NAD+ was not lower in the aging whole brain [42,43].

NAD+ is a precursor for the synthesis of NADP, which in its reduced form, NADPH, is essential for electron donation to a number of antioxidants such as glutathione and thioredoxin [44]. Both decreased availability of NAD+ and increased oxidative stress will therefore influence NADP+/NADPH ratios and NADPH levels. When we determined the NADP+/NADPH ratios in DJ-1 KO and wild type brains there were, in contrast to NAD+/NADH ratios no age-dependent reduction from 6 to 10 months of age (Figure 3B). The total amount of NADPH was significantly reduced in 24 months of DJ-1 KO brains compared to wild type (Figure 4B), possibly reflecting an increased amount of oxidative stress and reduced availability of its precursor.

### 2.4. Altered Behavior Preceeds Down-Regulation in Tyrosine Hydroxylase, but Co-Incides with Changes in the Mitochondrial Proteome

We have previously shown that tyrosine hydroxylase levels were not altered in the DJ-1 KO at the 3 months stage but were downregulated in the late adult stage [13]. Even at the 7 and 10 month stage we did not observe any significant changes in tyrosine hydroxylase levels (Appendix A). As we have previously observed mitochondrial changes in the total brain proteome at 3 months stage [13], we now isolated brain mitochondria and performed label-free proteome analysis from 12 months old adult stage at which changes in behavior and loss-of weight gain had occurred in DJ-1 KO (Figure 2). This gave us the opportunity to both anchor changes to the mitochondria and also go deeper into the proteome changes. We identified 5002 different proteins (Appendix A). Based on the criteria that the protein had to at least be detected in all WT or DJ-1 KO samples, 92 proteins were shown to be significantly regulated (Appendix A). In Table 1 we show the most regulated proteins and their suggested association with mitochondrial dysfunction and neuronal function.

The most up-regulated proteins in DJ-1 KO could be linked to oxidative stress and mitochondrial homeostasis regulation. Interestingly, also proteins associated with synapse signaling, and developmental disorder (methyl-CpG-binding protein 2) were shown to be regulated.

NADH dehydrogenase iron-sulfur protein 6 (NDUFS6), Receptor protein serine/threonine kinase (ACVR1l), and Integrin beta were only detected in WT samples (Table 1). NDUFS6 is required for mitochondrial complex I assembly and mutations in NDUFS6 are linked to adult-onset neurodegeneration. This is in line with previous research showing that DJ-1 associates to mitochondrial complex I as a response to oxidative stress [55] and that DJ-1 loss alters complex I activity [13]. ACVR1l and integrin beta are involved in the inflammatory response. ACVR1l function is involved in SMAD-signaling to dopaminergic neurons [53]. Amongst the most down-regulated proteins was EH-domain containing 1, a protein previously linked membrane trafficking and PD [47].

Down-regulated proteins included proteins with function in ER-stress response regulation, unfolded protein response and ubiquitin-proteasome dependent proteolysis. DJ-1 has been shown to regulate ER-mitochondria association and to be involved in unfolded protein response and ER-stress response [56,57,58]. Dysregulation of ER stress response affect mitochondrial dynamics [59]. Interestingly, knockout of the ER transmembrane protein Wfs1 results in a number of similar dysfunctions as we observe in the DJ-1 KO including retina thinning, loss of retinal ganglion cells, growth retardation, increased anxiety and movement defects [60,61,62]. Wfs1 acts upstream and DJ-1 downstream of PINK/Parkin [59,63], thus their signaling pathways converge and may be reflected in their deficiency-induced phenotypes. On the other hand, DJ-1 is also shown to work in parallel with the PINK/Parkin pathway [64], thus opening for the possibility that DJ-1 may affect mitochondrial function also through alternative actions. 

### 2.5. DJ-1 Deficient Zebrafish as a Model of Parkinson’s Disease

Even though DJ-1 associated recessive early-onset PD is rare [3], the role of DJ-1 in PD may be extended to idiopathic PD due to its susceptibility to oxidative modifications that render it irreversibly dysfunctional [65,66]. Thus, the DJ-1 PD model presented here likely reflects the effect of increased oxidative stress and its relevance for PD. Here we show that DJ-1 deficiency induces reduced locomotor activity and non-motor symptoms such as increased anxiety and freezing. The behavior defects were accompanied by changes in the mitochondrial proteome. Changes in NAD-metabolism seemed to be age-dependent and not correlated to behavior defects. On the other hand, we cannot exclude the possibility of a positive symptomatic effect of NAD-related treatment since DJ-1 KO had significantly lower levels of NAD and NADH at the latest adult stage compared to wild type animals. The characterization of the sequential events of molecular changes and phenotypic appearance may provide novel tools for understanding the cause and consequences in the development of PD and the effect of novel therapeutics.

## 3. Material and Methods

### 3.1. Animal Maintenance

Zebrafish were housed at the Zebrafish Facility located in the Department of Biological Sciences at the University of Bergen. The facility is run according to the European Convention for the Protection of Vertebrate Animals used for Experimental and Other Scientific Purposes. Adult zebrafish were maintained at 26–28 °C with a 14/10 light cycle and were fed twice daily. Embryos were maintained at 28 °C and raised in E3 buffer (5 mM NaCl, 0.17 mM KCl, and 0.33 mM MgSO_4_) until 14 days post-fertilization.

### 3.2. Zebrafish Lines

We have previously established a DJ-1 knockout (DJ-1 KO) line [13]. The line was established using CRISPR-Cas9 method by targeting the exon 1 of *park7* in Tübingen AB wild type (WT). The line was approved by the National Animal Research Authority at Mattilsynet (FOTS ID8039 and ID14039).

### 3.3. Weight Measurement

Adult male DJ-1 KO and WT (6, 12, and 18 months) animals were anesthetized in tricaine solution and weighed before being rehoused in the main system. Animals were from parallel tanks with the same number of fish.

### 3.4. Novel Tank Test

An in-house novel tank testing system was fabricated using a clear polypropylene tank of dimensions 30/20/10 cm. The behavior analysis was performed in an isolated room free from external noise. An electric heating pad was used underneath the tank to maintain the water temperature at 28 °C.

A CMOS DSLR (canon 5D mark III) was used to record the videos of the fish behavior. The camera was mounted at the front of the behavior testing tank.

Animals were acclimatized for 1 h in a pre-treatment tank in the recording room prior to tracking. Each animal was left and recorded for 7 min. The first 2 min were not considered for analysis. The tank water was changed after every recording to reduce the presence of pheromones which could potentially alter the behavior.

Both motoric and non-motoric parameters were analyzed. Motoric behavior was recorded as distance moved in 2D. Time spent in the lower level of the tank before making transition to the upper region of tank (Latency to upper half) and total absence of movement (freezing) were interpreted as anxiety and thus defined as non-motoric [67,68].

### 3.5. Zebrafish Behavior Analysis

A python-based particle tracking software was employed to analyze zebrafish behavior. First, the recorded color video was converted to a grey scale. To remove the tank and other static features, a base frame from all subsequent frames was subtracted leaving just the swimming fish against a black background. The base frame was chosen carefully so that the full trajectory was captured.

The trajectory was measured from frame to frame. The “Difference of Gaussians” approach was employed to locate all the features in each frame. Usually, there is only one feature, the fish, and in this case, the new position of the fish is added to the trajectory. In the rare case that there are no features detected in a frame, the last known position of the fish was added to the trajectory. Occasionally, more than one feature is detected. For example, if the fish is close to the bottom of the tank, its reflection can be visible in the glass. In this case, the feature closest to the last known position of the zebrafish was chosen and added to the trajectory. The final possibility is that a non-fish feature is detected, but the fish is not. To handle such a situation, a maximum frame-to-frame move distance for the fish of 5000 pixels was set. If this is exceeded, it was assumed that the detected feature is not the fish, and therefore the last known position to the trajectory was added.

The above algorithm requires that in the first frame, the only feature detected by the difference of the Gaussians method is the fish. Therefore, frames are dropped from the start of the video until this is the case. The above-mentioned setup was used to determine the distance travelled by the fish. The software was deposited at the following open access https://zenodo.org/record/7544764#.ZCLHGfZBxPZ (accessed on 17 January 2023).

### 3.6. Sampling of Adult Zebrafish Brains

Age-matched male DJ-1 KO and WT were anesthetized in tricaine solution and, upon the cessation of movement, euthanized by placing them in a 4 °C ice slush. The brains were then excised, snap-frozen in liquid nitrogen, and stored at −80 °C until further use.

### 3.7. Sample Preparation for Metabolomics Directed Mass Spectrometry and Protein Measurement

400 μL CH_3_CN:CH_3_OH: H_2_O (50:40:10, *v*:*v*:*v*) was added to each brain. The samples were then sonicated in cold water for 15 min in an Ultrasonic bath prior to vortexed and put on ice (5 s, 3 times). Lysate was thereafter centrifuged at −2 °C for 10 min (16,000× *g*) to precipitate protein. Supernatants were used for metabolomics analysis and pellets for normalization based on protein concentration. Both supernatant and pellets were stored at −80 °C.

Pellets used for protein concentration were evaporated with N_2_ and resuspended in homogenization buffer (10 mM K_2_HPO_4_, 10 mM KH_2_PO_4_, 1 mM EDTA, 0.6% CHAPS, 0.2M Na_3_VO_4_, 50 mM NaF, protease cocktail) (Roche) before sonication on ice for 20 s. The samples were then centrifuged at 13,000× *g* for 15 min at 4 °C and protein concentration was measured using NanoDrop 1000^®^ (Thermo Fisher Scientific, Mundelein, IL, USA).

### 3.8. Targeted Metabolomic Analysis

Accurate mass analysis was performed using a Thermo QExactive mass spectrometer (Thermo Fisher Scientific, Bremen, Germany) (HRMS) interfaced with the Dionex UltiMate 3000 liquid chromatography system (Thermo Fisher Scientific, Sunnyvale, CA, USA).

Liquid chromatography separation was performed using an Atlantis Premier BEH Z-HILIC VanGuard FIT Column (Waters, Selangor, Malaysia). The column compartment was kept at 25 °C during analysis and the total run time was 30 min. The injection volume was 10 μL, and the autosampler temperature was 5 °C.

Mobile phase A consisted of 30 mM ammonium carbonate in Milli-Q water (pH 7), whilst mobile phase B was acetonitrile. The flow rate was kept at 0.3 mL/min, and the following gradient was used: 78% of solvent B for 3 min decreased to 60% over 18 min, followed by 10 min of column equilibration with solvent B at 78%. Ions were monitored in positive Full MS and targeted single ion monitoring (t-SIM) mode. The Full MS scan range was 100–1000 *m*/*z* with a resolution of 17,500 at *m*/*z* = 200. T-SIM resolution was 70,000 at *m*/*z* = 200, isolation window of 8 *m*/*z*, with an inclusion parameter list (Table 2). Other MS parameters were sheath gas flow rate 48 (arbitrary units), aux gas flow rate 11 (arbitrary units), sweep gas flow rate 2 (arbitrary units), spray voltage 3.5 kV, capillary temperature 256 °C, S-lens RF level 30, AGC (automatic gain control) target 2E5, and maximum injection time 200 s. Exact mass acquisition and relative quantification of polar metabolites were performed with the XCalibur Quan Browser software (version 3.0.63) using a 5-ppm mass tolerance and referencing an in-house library of chemical standards.

### 3.9. Western Blotting

Protein pellets from 7 and 10 months brains were dissolved in homogenization buffer (10 mM K_2_HPO_4_, 10 mM KH_2_PO_4_, 1 mM EDTA, 0.6% CHAPS, 0.2 M Na_3_VO_4_, 50 mM NaF, protease cocktail (Roche, Basel, Switzerland)), separated by SDS-PAGE and transferred to PVDF membranes. The membranes were blocked for 1 h at RT followed by incubation with anti-tyrosine hydroxylase (MAB 318, Millipore, Burlington, MA, USA) or anti-actin for 1 h at RT. An appropriate secondary antibody was then used for 1 h at RT. Membranes were probed using Supersignal west pico plus Chemiluminescent substrate (Thermo Scientific, Waltham, MA, USA). Ponceau S staining was used as loading control.

### 3.10. Sample Preparation of Mitochondrial Lysates

Brain (12 months) were sampled as in 3.6 and directly proceeded to mitochondrial isolation. Two brains at a time were added to a Dounce homogenizer for lysing the cells with 7× tight and 7× loose pestles in a mitochondria isolation buffer (3 mM EDTA, 250 mM Sucrose, 100 mM HEPES pH 7.5). The homogenate was sonicated (2 × 5 s) and centrifuged twice at 1000× *g* for 10 min at 4 °C. Supernatant was then centrifuged at 7000× *g* for 10 min to pellet mitochondria. Mitochondrial pellet was dissolved in homogenization buffer and mitochondria lysed by sonication (4 × 5 s). Lysate was cleared at 15,000× *g* for 15 min.

### 3.11. Label Free Proteomics Analysis of Brain Mitochondria Fractions

Three individual mitochondria extracts for both WT and DJ-1 KO were analyzed by liquid chromatography tandem mass spectrometry (LC-MS/MS). The protein extracts (10 µg) were digested using the FASP method followed by reduction, alkylation, and peptide up-concentration as described in Frøyset et al., 2018 [16].

Peptides were separated using an Ultimate 3000 Rapid Separation LC system (Thermo Fischer Scientific, Sunnyvale, CA, USA). About 0.5 ug protein as tryptic peptides were dissolved in 2% acetonitrile (ACN), 0.5% formic acid (FA). The sample was loaded and desalted on a pre-column (Acclaim PepMap 100, 2 cm × 75 µm ID nanoViper column, packed with 3 µm C18 beads) at a flow rate of 5 µL/min for 5 min with 0.1% trifluoroacetic acid.

Separation was done on a 25 cm analytical column (PepMap RSLC, 25 cm × 75 µm ID. EASY-spray column, packed with 2 µm C18 beads) at flow rate of 200 nL/min. Solvent A and B were 0.1% FA (vol/vol) in water and 100% ACN respectively. Total run time was 195 min. The gradient composition was 5% B during trapping (5 min) followed by 5–7% B over 1 min, 7–16% B for the next 94 min, 16–32% B over 52 min, and 32–80% B over 3 min. Elution of very hydrophobic peptides and conditioning of the column were performed for 15 min isocratic elution with 80% B and 20 min isocratic conditioning with 5% B respectively. Total run time was 195 min.

The LC system was coupled to an Orbitrap Eclipse (Thermo Fischer Scientific, Bremen, Germany) mass spectrometer equipped with an electrospray ionization source (HESI).

Ions were monitored in positive DDA mode (data-dependent acquisition) with full-scan MS spectra range of 375–1500 *m*/*z* and resolution of 120,000 at 200 *m*/*z*. The most intense peptides (above 50,000 counts with charge states 2 to 5) were isolated in the C-trap before fragmentation (MS/MS). Data dependent ions were fragmented with a normalized collision energy of 30, and the resulting MS/MS spectrum was collected with a resolution of 15,000 at *m*/*z* 200. One MS/MS spectrum of the precursor ions was allowed before dynamic exclusion for 30 s with “exclude isotopes” on and 10 ppm of mass tolerance.

Data analysis was performed with Proteome Discoverer Version 2.5.0.400 (Thermo Fisher Scientific) and data were monitored with SequestHT and MS Amanda 2.0 (27 August 2021) using *Danio rerio* UniProtKB (reviewed_no_canonical_59162_160521.fasta) for the peptide matching. The results were filtered for high confident peptides and signals were normalized and scaled to the same total peptide amount per channel (average abundance per protein and peptide was 100).

A maximum of two missed trypsin cleavages were allowed with 10 ppm tolerance. In order to be used for quantitation, only proteins identified with at least two unique peptides were considered. The protein list was further reduced by removing potential contaminants and proteins that had not been identified in all three DJ-1 KO or WT samples. Intensity values were considered significant if the passed a one sample *t* test (DJ-1 KO and WT only) or a two-sample *t* test (proteins found in both DJ-1 KO and WT samples).

### 3.12. Statistics

Data analyses were performed using Graphpad Prism 9.5.0 and either Wilcoxon or two-way ANOVA tests as stated in the specific figures. For NAD+/NADH and NADP+/NADPH, normality and lognormality testing in Prism indicated that the data were likely to have been sampled from lognormal distributions. Significance testing was therefore performed on Log10- transformed data, using two-way ANOVA with Tukey multiple comparisons test.

## Figures and Tables

**Figure 1 ijms-24-06456-f001:**
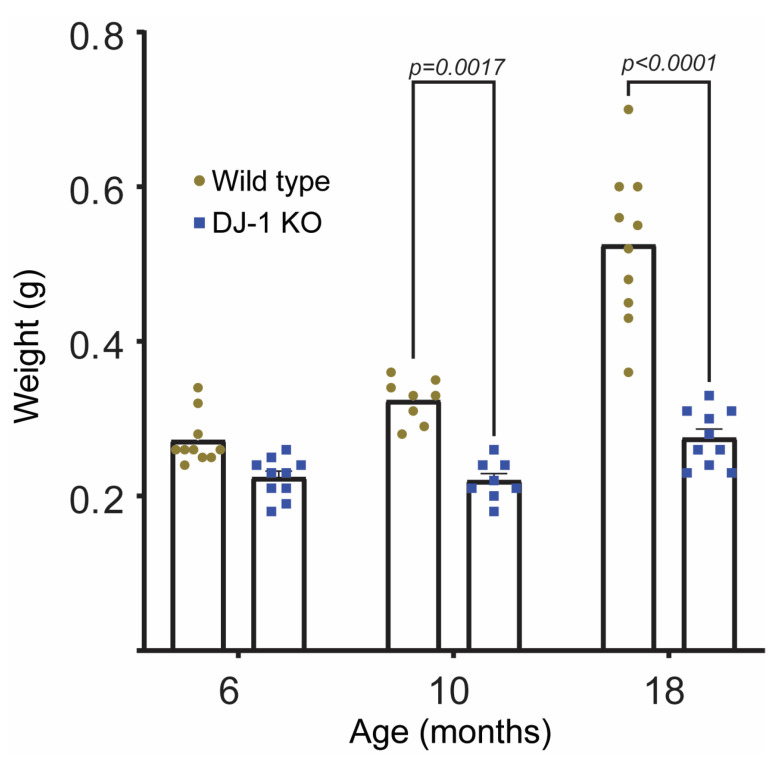
DJ-1 knockout fish fail to gain weight as compared to their wild type littermates. From 6 to 18 months of age DJ-1 KO do not gain wait in contrast to wild type animals. Data are the mean ± SEM (*n* = 8–10). Using two-way ANOVA a significant interaction between genotype and age was shown (F (2,50) = 21.52, *p* < 0.0001). *p* values represented in the figure were obtained using Tukey’s post hoc analysis.

**Figure 2 ijms-24-06456-f002:**
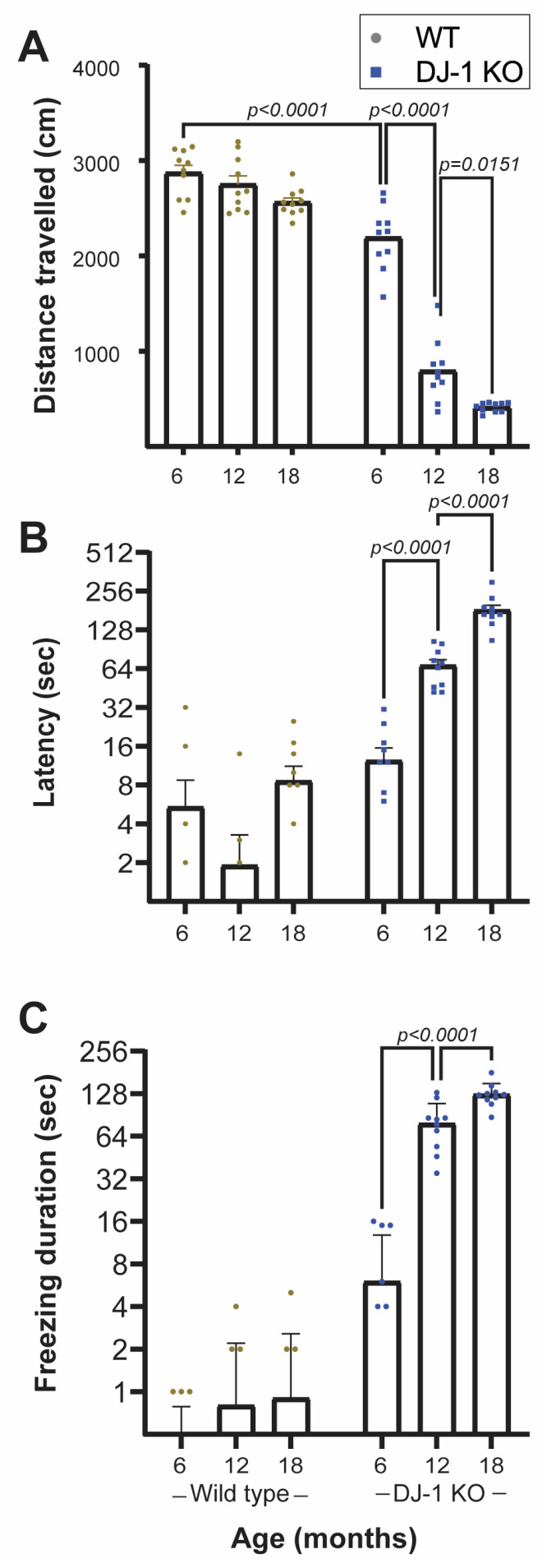
Behavior analysis. Using two-way ANOVA a significant interaction between genotype and age was shown in regard to total distance traveled (F (2,54) = 44.43, *p* < 0.0001) (**A**), latency to reach upper half of the test tank (F (2,54) = 60.99, *p* < 0.0001) (**B**) and freezing duration (F (2,54) = 71.03, *p* < 0.0001) (**C**). Data represent the mean ± SEM (*n* = 10). *p* values were obtained by Tukey’s post hoc analysis.

**Figure 3 ijms-24-06456-f003:**
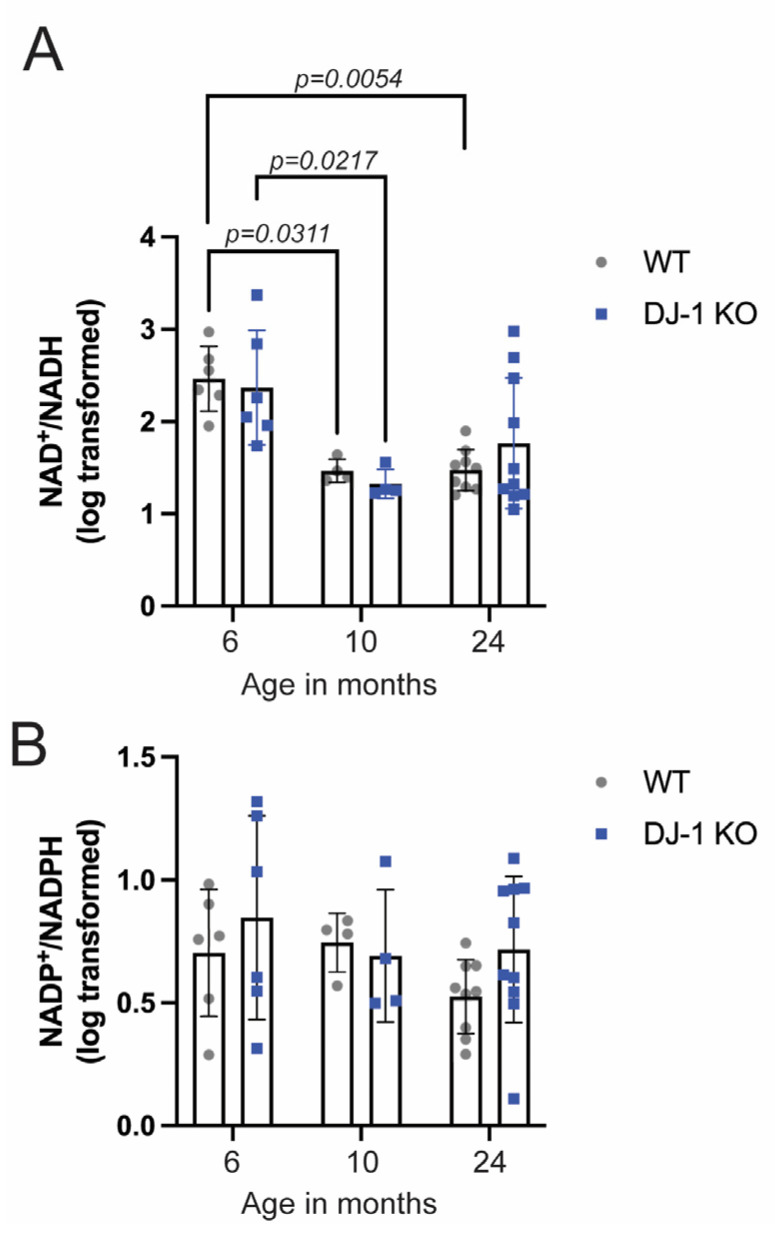
NAD metabolism in the aging DJ-1 knockout and wild type zebrafish brains. A two-way ANOVA revealed that there was no significant interaction between age and genotype (F (2,33) = 0.80, *p* = 0.4297) in NAD+/NADH ratio. A significant effect was only seen in relation to age (F (2,33) = 14, *p* < 0.0001). Data represent the mean ± SD. *n* = 6 (6 months), *n* = 4 (10 months), *n* = 9 (24 months). *p* values are obtained with Tukey multiple comparisons test on Log10 transformed data.

**Figure 4 ijms-24-06456-f004:**
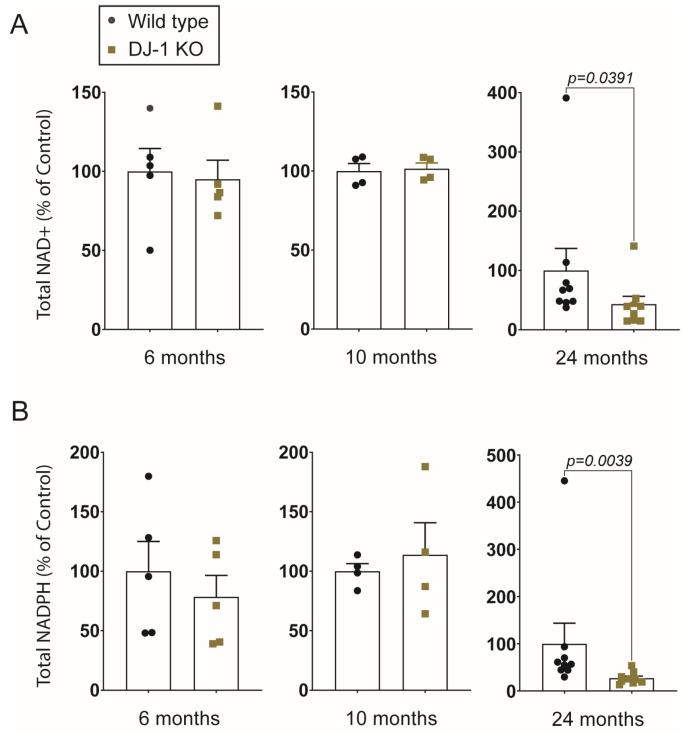
Relative amount of NAD+ and NADPH in DJ-1 knockout brains compared to wild type. Mass spectrometry measured areas of NAD+ (**A**) and NADPH (**B**) of individual brains were normalized to the protein content for each age group. Values are the mean ± SEM. *p* values are from Wilcoxon test.

**Table 1 ijms-24-06456-t001:** Regulated proteins brain mitochondrial fractions from 12 months *park7−/−* animals.

UniProt Acc ^a^	Protein Names	Gene Name	*p*-Value	F.C ^b^	KO ^c^	WT ^d^	Protein Description	
**Transport and membrane trafficking**								
A9JRA5	Gelsolin	*scinla*	0.0073	4.62	22.69	20.50	Linked to OXPHOS defect [45].	
Q6NZW3	Cofilin 2	*cfl2*	0.0482	4.47	20.54	17.87	Mitochondrial homeostasis [46].	
Q66HW2	EH-domain containing 1	*ehd1b*	0.0419	*0.45*	19.90	21.11	Downregulated in PD [47].	
X1WC49	Anoctamin	*ano3*	0.0204	*0.32*	19.99	20.94		
**Synapse signaling**								
A2CJ03	Dystrobrevin	*dtna*	0.0003	5.20	19.47	17.23		
A5WWH0	Chromogranin B	*chgb*	0.0078	3.74	21.13	19.05		
**Metabolism**								
Q1RLP8	Suppressor of tumorigenicity 14 protein homolog	*st14a*	0.0285	8.89	20.45	16.89	Mitochondrial localization stress response [48].	
Q6P962	Glutathione transferase	*gsta.1*	0.0081	3.70	24.74	22.62	Oxidative stress regulation	
Q66I52	NADH dehydrogenase iron-sulfur protein 6	*ndufs6*	n.a.	*n.a.*	NaN	25.20	Complex I assembly	
A0A0R4IQ88	D-glutamate cyclase	*dglucy*	0.0012	*0.27*	22.68	24.50	Metabolization od D-glutamate	
Q7ZU10	Aldehyde dehydrogenase	*aldh3b1*	0.0135	*0.47*	17.71	18.61		
B3DJF3	3′-phosphoadenosine-5′-phosphosulfate synthase	*papss1*	0.0004	*0.49*	20.44	21.22		
**Chaperone activity**								
A0A2R8RNM0	Mitochondrial import inner membrane translocase	*timm8a*	0.0249	3.37	20.11	18.19		
**Mitochondrial transcriptional and translational regulation**								Oxidative stress regulation [49].
Q7T2T7	Methyl-CpG-binding protein 2	*mecp2*	0.0417	6.71	21.22	18.47	Ass. to neurologic developmental disorder	
Q502J9	39S ribosomal protein L30, mitochondrial	*mrpl30*	0.0266	*0.14*	19.09	21.94		
E9QEK4	Mitochondrial ribosomal protein L45	*mrpl45*	0.0158	*0.36*	21.05	22.11		
A2BEV1	Mitochondrial translational initiation factor 3	*mtif3*	0.0020	*0.36*	19.51	21.04		
**Stress response regulation and unfolded protein response**								Required for ER stress respons.
A8WG75	Pdia5 protein	*pdia5*	0.0019	*0.05*	20.91	24.84	ER stress regulation [50]	
Q7ZUW0	DnaJ (Hsp40) homolog, subfamily A, member 3A	*dnaja3a*	0.0089	*0.47*	21.19	21.72	Reg. of mitochondrial apoptosis signaling	
U3N8Z0	Transient receptor potential cation channel memb. 3	*trpc3*	0.0071	*0.48*	18.53	19.44	Mitochondrial Ca2+ control [51]	
**Ubiquitin-Proteasome Dependent Proteolysis**								
E7EZD6	Ubiquitinyl hydrolase 1	*usp4*	0.0165	*0.21*	20.09	21.63		
Q9PTH5	Proteasome activator subunit 1	*psme1*	0.0368	*0.22*	20.86	21.78		
**Mitochondrial quality control**								Linked to PD [52].
B0R198	Voltage-dependent anion-selective channel protein 3	*vdac3*	0.0433	*0.39*	21.37	22.45		
A0A0R4IXA9	Adenylyl cyclase-associated protein	*cap2*	0.0099	*0.43*	20.30	21.44		
**Neuroinflammation**								Neuroprotective. Linked to early loss of axons in PD models [53].
O73736	Receptor protein serine/threonine kinase	*acvr1l*	n.a.	*n.a.*	NaN	21.29	Regulation of mitochondrial function [54].	
Q3YA99	Integrin beta	*itgb1b.1*	n.a.	*n.a.*	NaN	18.90		
**Other**								
F1QYR3	Uncharacterized protein	*LOC100004199*	0.0425	*0.28*	18.62	20.35		
A0A286YA42	Si:ch73-366l1.5	*si:ch73-366l1.5*	0.0174	*0.19*	20.67	22.83		
A5PMM7	Si:dkey-193c22.1	*si:dkey-193c22.1*	0.0355	*0.38*	17.95	19.25		

^a^ UniProtKB release 2023. ^b^ Fold change DJ-KO/WT. ^c,d^ Mean abundance (log2 values). NaN: not detected. Unless referred to, information of protein biological and molecular functions is found through UniProt KB 2023.

**Table 2 ijms-24-06456-t002:** Mass standards used for calibration.

Compound	Accurate Mass	Formula [M + H]^+^	Inclusion List	rt
NAD^+^	664.11695	C_21_H_28_N_7_O_14_P_2_	665.0 ± 8	11.02
NADH	666.13205	C_21_H_30_N_7_O_14_P_2_	665.0 ± 8	5.57
NADP^+^	744.08273	C_21_H_29_N_7_O_17_P_3_	745.0 ± 8	10.54
NADPH	746.09783	C_21_H_31_N_7_O_17_P_3_	745.0 ± 8	10.57

## Data Availability

The data presented in this study are available in Appendix A.

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
