# Peer review of "Progressive Motor and Non-Motor Symptoms in Park7 Knockout Zebrafish"

_ijms, 2023, doi:10.3390/ijms24076456_

Round 1

Reviewer 1 Report

The manuscript by Chavali et al., describes the effects of DJ-1 KO in zebrafish as model reproducing PD. Specifically, the authors analysed PD-related motor and non-motor symptoms from an age-point of view. They also compare the NAD-metabolism in the aging brains of DJ-1 knockout and wild type animals and demonstrate an age-dependent decrease in NAD+/NADH ratios in both knockout and wild type, with a difference in NAD+ and NADPH levels first appearing at the late adult stage.

The manuscript is interesting and original in its topic because it proposes a novel handling model to investigate PD pathogenesis in terms of causes and consequences in the development of the disease and the effect of novel therapeutics.

However, there are further experiments that should be performed in order to strength the relevance of the model. 

1. The expression of TH, Dopamine transporter (DAT) and synaptic proteins should be evaluated at the different ages of the animal model, both in the WT and in DJ-1 KO in order to identify the time points of dopaminergic degeneration and synaptic dysfunction. These experiments will allow to correlate the biochemical parameters with the functional aspect of PD like motor symptoms. Moreover, these experiments might be useful also to discriminate the non motor symptoms appearing in the early stages of the disease thus allowing to define potential biochemical markers of the early stages of PD. All these experiments will make the model more appropriate to investigate the molecular mechanisms of PD and to explore the activity of novel pharmacological strategies

2. Experiments aimed to investigate the expression of further proteins involved in mitochondrial metabolism like MnSOD, NADPH-oxidase, GPH might be useful to better characterize the aspect related to mitochondrial dysfunction in DJ-1 Ko mice.

Author Response

We have previously shown a downregulation of TH levels at 16 months stage (Edson et al 2019). Also, in Edson et al we showed that there was no downregulation in TH levels at 3 mnth KO. We have now made new TH-blots for 10 mnths WT and DJ-1 KO and show that behaviour effects occur prior to TH downregulation.

We have included a label-free proteomics analysis of brain mitochondria (1 year). Data are shown as a new Table 1 with the most up-/down-regulated proteins in the DJ-1KO. We have also included two supplementary tables (S1 and S2) that shows all proteins identified and those that are found regulated. We have included methods for these data and also included relevant data in the rest of the manuscript.

Reviewer 2 Report

This is an interesting manuscript with valuable data. DJ-1 or PARK7 is an important gene in the context of Parkinson's disease. PARK7 function is largely unknown, therefore, this paper provides very important information about its role. I have several comments taht authors can easily address:

1. Nomenclature, I think PARK7 or park7 for Danio rerio, is an official symbol of the gene.  DJ-1 is conventional or historical name. It is ok to use it, but maybe to increase to visibility and to make the paper to be better to find, official term "park7" should be preferred. Park7 is used more often and makes the article easier to find later.

2. Knockout technology. Knockout mice are well known for their "footprint effect" that was described in these two papers (PMID: 17331107 and 19293327). As zebrafish strains are also inbred, that could significantly impact the results. I think this is important for authors to discuss and consider in their discussion.

3. Interestingly, one main trait in KO fish is the growth retardation. This is quite similar the mouse knockout of WFS1 gene (PMID: 19293327). WFS1 gene knockouts have growth retardation, increased ER stress and mitochondrial problems (PMID: 30584460). WFS1 gene is linked to mitochondrial functions and neurodegeneration (PMID: 23321269).  Possibly authors can somehow overlay the potential functions of Park7 and WFS1 genes. Both of these genes seem to have cellular protective effect and they have connection with the neurodegeneration.

4. The same WFS1 gene is linked to mood disorders and anxiety, deletion of this gene in mice induces increased anxiety (PMID: 19041897) and it is related to mood disorders in humans (PMID: 15473915). This seems to be significant overlap with the Park7 and WFS1 effects.

5. Moreover, WFS1 deficient mice have shown to have impaired dopamine release in striatum (PMID: 20972658), the very structure linked to PD.

6. Authors focus mostly on the mitochondrial function and this is very much understandable in the context of PD. However, they should also, at least discuss, more the role of Park7 in the ER stress or unfolded protein response. This seems to the reviewer that Park7 deletion has very large overlap with WFS1 deletion and this indicates that ER stress could be the primary mechanisms behind the phenotype. Park7 is a redox sensitive chaperone, WFS1 is also chaperone and its over-expression was shown to reduce the neurodegeneration and tau-pathology in mouse models (PMID: 35389045). Park7 and WFS1 seem to have inherent overlap in their functions. And this should be discussed.

Author Response

We have changed the title which now include park7 not DJ-1. We have also included park7 in key words.

The lines investigated were only backcrossed once (F5), but we do have a line in which

DJ-1 is expressed in glia cells in the DJ-1 null background (Gharbi et al 2021). Unfortunately we do not have behavior data for this line, and with a 10 days response time it is impossible to obtain. BUT, we do have weight data comparing 18 months WT, DJ-1 KO, and DJ-1 KO with reinsertion of DJ-1 in glia cells:

WT: 0.45g+/-0.13, DJ-1 KO: 0.27g+/-0.03, glialDJ-1:0.36g+/-0.09 showing a significant reduction in weight of DJ-1 KO compared to both WT and glialDJ-1. In the Gharbi et al 2021 we also show that reinsertion of glial DJ-1 rescues DJ-1 retinal phenotype of DJ-1 KO.

There is a published line of Wfs1 deficient zebrafish line Cairns et al 2021. In contrast to the DJ-1 knockout this line has severe development effects. Also, Wfs1 and park7 is not located on the same chromosome.

But, indeed there is a high resembles between Wfs1 and DJ-1 knockout animals incl. effect on retina thinning and ganglion cells in zebrafish and behavior in mouse models. We searched for Wfs1 in all our proteome data sets in order to determine any possible co-regulation, but neither in the 5 dpf larvae or in the adult brain proteomes (regardless of knockout or not) Wfs1 was detected. Both, datasets include more than 7000 proteins. On the other hands, in the mitochondrial proteomics analysis which we now have included (Table 1 and supplementary tables) we do indeed see a regulation in UPS and ER stress associated proteins. Thus, even though these two genes may not be directly interconnected their neuroprotective pathways may indeed converge. We have added the possibility of ER stress involvement and converging pathways in the revised manuscript.

Reviewer 3 Report

The manuscript by Fladmark and Colleagues begins to further characterize their DJ-1 KO zebrafish model that was previously published by measuring locomotion at different ages along with levels and ratios of metabolites such as NAD+, NADPH and the ratios of NAD+/NADH and NAD+/NADPH. In addition, they measure what they define as non-motoric behaviors such as freezing that have been previously associated with more anxiogenic phenotypes. While additional genetic models, in particular non-rodent models, of PD are important, the study has some major limitations and the conclusions are not supported by the data as presented.

1.     They state “as the DJ-1 knockout zebrafish also shows non-motor symptoms such as  increased anxiety and dysregulation in NAD-metabolism. . .”

a.     The link between freezing and anxiety, while cited, may not hold in this context as you already have motor deficits in the zebrafish at 6 months of age. How can you disambiguate the motor changes from the anxiety measures as while these are non-motoric form the citations, it does not mean they are not being driven by the lack of movement from the fish.

b.     The dysregulation in NAD ratios is only age-dependent and not genotype-dependent. However, the decrease in NAD+ and NADPH occurs in DJ-1 KOs, but only at 24 months. Therefore, it seems that these changes neither correlate with, nor are responsible for, movement deficits. Moreover, the statement “At 24 months of age NADP+/NADPH in DJ-1 KO seemed to be somewhat increased” is not backed up by the variability and overlap of the error bars with the mean. This, in addition to a lack of TH deficits at young ages in their previous work, suggest that mechanistic insight into motor deficits is lacking in this model and are not the same as what is observed in humans wherein a loss of ~85% of dopamine terminals within the striatum associates with motoric deficits. This would be fine, but there is no alternative explanation for the motor changes nor for what are described as anxiety changes.

2.     Statistics

a.     These are not done correctly. None of the overarching Two-WAY ANOVA values are given (e.g. Age, Genotype, Interaction). It appears that only a post-hoc analysis is shown and sometimes that post-hoc analysis should not be performed (for instance comparing WT 6-month to DJ-1 10 month KO if there is no interaction term).

3.     Figure 4

a.     Why are the individual data points not shown here. I understand it is normalized to the control, but it is not clear if the variability in each group is from technical replicates or biological replicates. Also, if using a standard, why can’t you obtain total amounts per µg of protein extracted or something to that effect rather than doing percent control. More details on the N-values for this figure are required.

4.     Minor errors

a.     Line 51, has = have

b.     Line 58, genetic = genetically

c.     Line 72, an = a

d.     Line 102, wild types = wild type fish

e.     Line 108, deficits such as …

f.      Line 192, it = its

g.     Line 240, euthanized = anesthetized

h.     Line 294, evaporation = evaporated

Author Response

We have included (supplementary files) representative recordings of fish (WT and KO) in their natural environment and novel tank to show that the KO fish enter the upper half of the tank as WT when they are in their normal environment. Supplementary files are referred to in the text.

“At 24 months of age NADP+/NADPH in DJ-1 KO seemed to be somewhat increased” is deleted. Should have read NAD+/NADPH, but can be taken out. The absence of correlation between behavior and NAD-metabolism is now highlighted.

We have added TH blot and an analysis of the mitochondrial proteome at 1 yr stage to show that mitochondrial linked effects may be the driving factor for behavior changes.

We have removed wrong bracket and done some editing of the related text.

Figure 4 has been changed and now show individual fish (biological replicates). Our MS-standards are only used for pick finding for the LC-MS and not quantitation-this because we do not have the ability to incorporate labelled metabolites in the fish in contrast to what would be possible in cell culture. Also, as stated in the manuscript samples with different ages were now analyzed on the LC-MS in a single run, thus injection volume may have variation. Only ratios are therefore shown. Normalization is always a problem since the reproducibility of protein concentration measurement varies and in our case was only done in each age group at the same time.

Spelling mistakes have been corrected

Round 2

Reviewer 3 Report

They mention a tyrosine hyrdoxylase blot in the response and have information about it in the methods, but I did not see it in the supplementary files or the main text. Maybe was lost on upload or something happened? I would like to see that. In addition, the author's still have not added the actual ANOVA values for their multiple comparisons. They show the post-hoc test, but the F vaules and p-values for the ANOVAs must be reported. Finally, the font changed when some of the corrections were made. Make sure that all font is consistent upon proofing.

Author Response

We have now included a figure (S5) with TH-expression from 7 and 12 months brain samples.

We have added ANOVA values in either figure legends, main text or both.

In addition:

We have also added text in 2.4 that somehow was lost in the previous version.

We have re-arranged the numbering of the Supplements according to the journal’s requirements.

Round 3

Reviewer 3 Report

The authors have fully addressed my critiques/concerns